# The Construct CoSe_2_ on Carbon Nanosheets as High Sensitivity Catalysts for Electro-Catalytic Oxidation of Glucose

**DOI:** 10.3390/nano12030572

**Published:** 2022-02-07

**Authors:** Di Wang, Ying Chang

**Affiliations:** 1Inner Mongolia Key Laboratory of Green Catalysis and Inner Mongolia Collaborative Innovation Center for Water Environment Safety, College of Chemistry and Environmental Science, Inner Mongolia Normal University, Hohhot 010022, China; 2120200811@mail.nankai.edu.cn; 2Key Laboratory of Advanced Energy Materials Chemistry (Ministry of Education), Renewable Energy Conversion and Storage Center (RECAST), College of Chemistry, Nankai University, Tianjin 300071, China

**Keywords:** selenide cobalt, glucose, electric catalysis oxidation, carbon nanomaterials

## Abstract

Seeking an efficient, sensitive, and stable catalyst is crucial for no-enzyme glucose sen-sors to detect glucose content accurately. Herein, we constructed a catalyst of selenide cobalt (CoSe_2_) on carbon nanomaterials by auxiliary pyrolysis of sodium chloride. The CoSe_2_ on carbon nanosheets possesses good selectivity and a wide linear range up to 5 mM. Based on its good detection per-formance, the CoSe_2_ nanomaterial is expected to be an emerging catalyst for no-enzyme sensors.

## 1. Introduction

Glucose, as an important carbohydrate in organisms, directly involves in all kinds of metabolism [1,2]. Therefore, it is crucial to keep a suitable concentration of glucose in human body [3,4]. As far as we know, diabetes has become the third-largest multiple disease in society, and the number of patients shows a rapid growth trend; thus, precisely and quickly detecting the glucose concentrations is of great significance [5,6]. Electrochemical glucose biosensors have drawn enormous attention with the advantages of simple operation, low detection line, and wide linear range. Therefore, glucose sensors exhibited broad application prospects in the fields of biological, clinical diagnosis, medicine, environmental protection, and military medicine. According to the different surface materials of modified electrode, glucose sensors can be divided into two types: enzymatic glucose sensors and non-enzymatic glucose sensors [7,8].

As we all known, enzymatic glucose sensors possess excellent selectivity and specificity, but they have deficiencies in the reaction process. For instance, they are negatively affected by the environment (temperature, pH, and humidity), their prices are expensive, and the enzyme cannot be reused; all of these things restrict their application [9,10]. Compared with enzymatic glucose sensors, the non-enzymatic one shows wide application prospects, with the advantages of good reproducibility, stability, and low price [11].

For high-performance glucose sensors, electrode materials play a vital role. In particular, nanomaterials exhibit many active sites and a large specific surface area, so they are promising materials in enzyme-free glucose sensors [12]. Noble metal nanomaterials, carbon nanomaterials, and transition metal-oxide nanomaterials are three kinds of common electrode materials for nonenzymatic glucose sensors. Gold (Au), platinum (Pt), etc., noble metal materials are the ideal catalysts because of their unique catalytic performance and biocompatibility, but they are easily inactivated by chloride ions, and the price is expensive [13,14,15,16,17]. Carbon nanomaterials, represented by graphene and carbon nanotubes, possess many advantages, such as a large specific surface area, high porosity, and good conductivity, significantly improving the detection performance. Recently, transition metal materials, such as copper (Cu), cobalt (Co), and nickel (Ni), have drawn more and more attention in regard to their merits, i.e., a significantly lower price than noble metal materials, strong stability, and good reproducibility. Additionally, they can avoid the electrode poisoning that is due to the electrons-transfer mechanism in the process of oxidation reduction [18,19,20,21,22,23]. Moreover, there are some studies that have reported that selenides have great potential in improving catalytic performance because of the increase on the covalence increase [24,25,26].

In this work, we used Co-Se carbon nanocomposites to detect glucose content in enzyme-free glucose sensors. Studies have suggested that, compared with Co single metal oxides, binary metal oxides often showed higher conductivity and electrochemical activity. Therefore, we fabricated Co nanoparticles first, and then selenylation in Ar atmosphere by sodium chloride (NaCl) assisted pyrolysis, reducing the aggregation of nanoparticles and increasing the active area and conductivity. The Co-Se carbon nanocomposites that we prepared showed excellent performance in the detection of enzyme-free glucose sensors.

## 2. Experimental Section

### 2.1. Materials

Ascorbic acid (AA), uric acid (UA), and dopamine (DA) were obtained from Sigma-Aldrich. Cobalt chloride hexahydrate (CoCl_2_·6H_2_O) was obtained from Shanghai Titan Chemical Co., Ltd. Glucose anhydrouse was obtained from Tianjin Fuchen Chemical Reagents Factory. D-(+)-sucrose, mannose, maltose, and fructose were purchased from Damas-beta.

### 2.2. Synthesis of Co-Se Carbon Nanomaterials

The synthesis schematic of nanocomposites is shown in Figure 1. At first, 0.714 g (0.06 M) CoCl_2_·6H_2_O, 0.5 g (0.06 M) anhydrous glucose, and 3.51 g NaCl (1.2 M) were dissolved in 50 mL deionized water to obtain a pink solution and then dried for 24 h in the freeze dryer. The purple product was then pyrolyzed at 800 °C for 3 h (5 °C/min) under Ar atmosphere. After washing the black product and drying at 60 °C, we mixed the sample with dicyandiamine for a 1:1 mass ratio. Then the mixture was continuously pyrolyzed at 800 °C for 3 h (5 °C/min) and at 300 °C for 2 h (2 °C/min). Afterward, the composite was selenited in an argon environment with a mass ratio of 1: 5 (the latter was the mass of selenium powder). According to the same conditions, we baked the products to 350, 400, 450, 500, and 550 °C, respectively, denoted as CoSe_2_-350, CoSe_2_-400, CoSe_2_-450, CoSe_2_-500, and CoSe_2_-550.

### 2.3. Characterization

The composition of the catalyst was investigated by using X-ray diffraction (XRD). X-ray photoelectron spectroscopy (XPS) was conducted to determine the element composition and valence state of the material. The morphology of the sample was observed by using scanning electron microscopy (SEM). Transmission electron microscopy (TEM) was used to characterize the lattice structure.

### 2.4. Electrochemical Measurement

All the electrochemical tests were carried out on a CHI760e electrochemical workstation, at room temperature. A standard three-electrode system was used: glass carbon electrode coated with prepared materials as working electrode, Ag/AgCl (saturated potassium chloride solution) as reference electrode, and Pt network electrode as counter electrode. The detection of catalyst performance was carried out in 0.1 M KOH electrolyte solution, using cyclic voltammetry (CV) and chronoamperometry (I–t) tests.

Working electrode preparation: mixed 2.5 mg sample, 225 μL water, 25 μL nafion solution, and 250 μL ethanol; then added 2.5 mg carbon powder; and performed ultrasonic for 30 min to mix evenly. Pipetted 6 μL (0.02 M) slurry onto the electrode surface, waited for it to dry so that it would not easily to fall off. The geometric area of the electrode is 0.07065 cm^2^.

## 3. Results and Discussion

The XRD patterns of samples at various temperature are shown in Figure 2, exhibiting that the diffraction peaks at 30.5, 34.2, 37.6, 43.7, 46.4, 51.8, 56.5, 58.8, 63.4, 74.0, and 76.0° correspond to the (200), (210), (211), (220), (311), (230), (321), (400), (421), and (332) planes of the cubic CoSe_2_ (PDF#09-0234) [27]. This result indicates that CoSe_2_ was successfully synthesized in different situations.

Figure 3a shows the full spectrum of XPS for Co-Se carbon nanocomposites, thus indicating the existence of Se 3d, Co 2p, N 1s, O 1s, and C 1s peaks. As shown in Figure 3b, the binding energies at 54.2, 55.8, and 58.7 eV confirm the existence of Se 3d_1/2_, Se 3d_5/2_, and the surface oxide of Se. The C 1s spectrum is shown in Figure 3c, with the peaks of binding energies at 284.2 eV (graphitic C), 285.6 eV (C in C–O bonds), and 288.2 eV (C in C=O bonds), respectively. The Co 2p spectrum is given in Figure 3d: the peaks at 778.4 and 781.6 eV indicate the existence of Co 2p_3/2_, the peaks at 793.5 and 797.1 eV are corresponding to the Co 2p_1/2_, and the peaks at 781.6 and 797.1 eV indicate the existence of surface oxides [27].

Figure 4a,b shows the SEM images of CoSe_2_-400; we can clearly observe that the nanostructures are mainly displayed as a sheet structure. TEM diagrams of the nanosheet are shown in Figure 4c. The element maps illustrate that the CoSe_2_-400 nanomaterial mainly comprises Co, Se, and C elements (Figure 4d), which are highly dispersed across the whole structure. In Appendix A, it can be clearly observed that the Co element accounts for 2.2% and the Se element accounts for 13.18% in the nanocomposite. The lattice fringes of the sample particles are shown in Figure 4e; the lattice spacing is about 1.595 nm, corresponding to the (023) crystal face of CoSe_2_. Moreover, we can observe the multiple diffraction rings from the selected area electron diffraction (SAED) images (Figure 4f), and this means that the CoSe_2_-400 has a certain crystallinity.

The electrochemical test was performed by cyclic voltammetry (CV). We set the scanning potential range of 0~0.7 V (vs. Ag/AgCl) in 0.1 M KOH solution with 7.0 mM glucose. As shown in Figure 5a, they are different for the response degrees of 350–550 °C samples to current density. Among these samples, we chose the CoSe_2_-400 sample as our research object to study the catalyst performance, due to it having the best response performance. Figure 5b shows the CV curves of CoSe_2_-400 carbon nanocomposites in the presence or absence of glucose. The peak current is significantly enhanced after adding glucose, thus indicating that the CoSe_2_-400 has a good response to glucose concentration. The relationship between the potential and the scanning rate is shown in Figure 5c, with an increase in scanning rate from 10 to 100 mV·s^−1^, a slight positive shift of the oxidation peak, and a slight negative shift of the reduction peak, all which can be observed. In Figure 5d, the peak current density is proportional to the square root of the scanning rate, indicating that the CoSe_2_-400 electrode in the KOH solution presents the diffusion control process.

The current versus time (I–t) curves are shown in Figure 6a; they were carried out at 0.55 V applied voltages, while introducing 0.5 M glucose into 0.1 M KOH electrolyte. As described above, the CoSe_2_-400 sample shows the best performance. Many factors affect the detection of the glucose sensor, including the selection of appropriate applied voltage. In order to set the optimal working potential in glucose detection, the I–t curves of CoSe_2_-400 were measured under various voltages of 0.45, 0.50, 0.55, and 0.60 V, respectively. Subsequently, as shown in Figure 6b, the current intensity at 0.55 and 0.60 V are relatively stronger. In order to get a high response current after the background current, we finally chose 0.55 V as the working voltage to make the sensitivity of glucose detection higher (Appendix A). Then, it can be found in Figure 6c that, as the glucose concentration increases, the current increases markedly. Thus, the CoSe_2_-400 has a good response for glucose detection. Figure 6d shows the current-versus-concentration (I–c) fitting curve corresponding to Figure 6c, whose linear range can be up to 5 mmol·L^−1^.

Selectivity is another factor in the detection of the glucose sensor. There are many substances in the human body, such as ascorbic acid (AA), dopamine (DA), uric acid (UA), etc., which influence glucose detection. Generally, at a normal physiological level, the content of these interfering substances is far lower than that of glucose. However, we further conducted the anti-interference experiments on CoSe_2_-400 to study whether the substances play an impact on detection of glucose. As shown in Figure 7a, 0.1 mM DA, AA, UA, and 1 mM glucose were added, in turn, at 0.55 V. Apparently, the current response caused by three kinds of distractors showed a negligible change compared with the response of adding glucose, illustrating that the effects of DA, AA, and UA on the detection of glucose could be ignored. Similarly, by adding 0.1 mM maltose, sucrose, mannose, fructose, and 1 mM glucose to the system, it can be observed that these sugars also have little effect on the detection of glucose and could be ignored (Figure 7b). Consequently, the electrode modified by CoSe_2_-400 has good selectivity for glucose at 0.55 V, and other substances can be ignored at the physiological level.

## 4. Conclusions

Overall, CoSe_2_ nanomaterials under different temperatures were successfully fabricated by sodium chloride–assisted pyrolysis. The results revealed that the catalyst at 400 °C showed the best performance, which possessed a wide detection range, good selectivity, and good detection response to glucose. Therefore, the CoSe_2_-400 nanocomposite is a promising material for enzyme-free glucose sensor.

## Figures and Tables

**Figure 1 nanomaterials-12-00572-f001:**
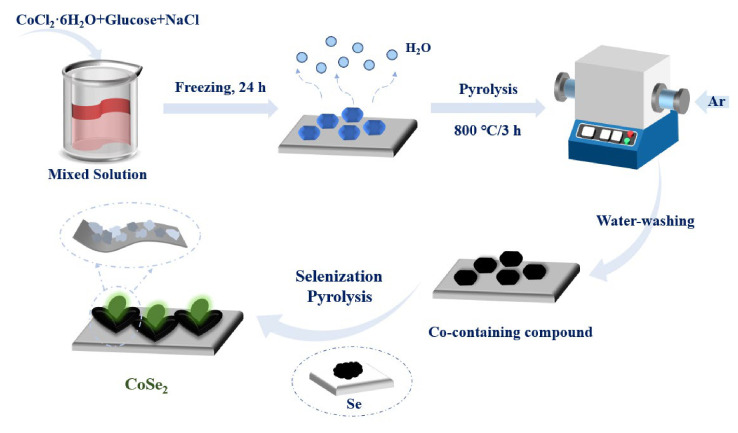
Synthesis diagram of CoSe_2_ nanocomposites.

**Figure 2 nanomaterials-12-00572-f002:**
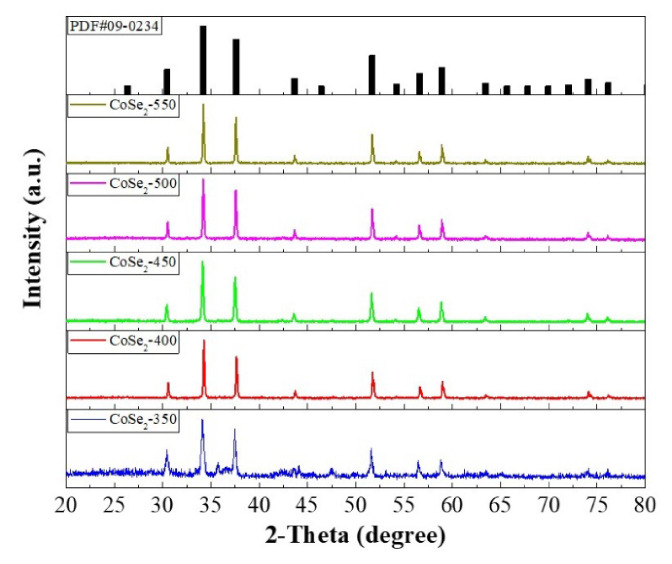
XRD patterns of CoSe_2_-350, CoSe_2_-400, CoSe_2_-450, CoSe_2_-500, and CoSe_2_-550.

**Figure 3 nanomaterials-12-00572-f003:**
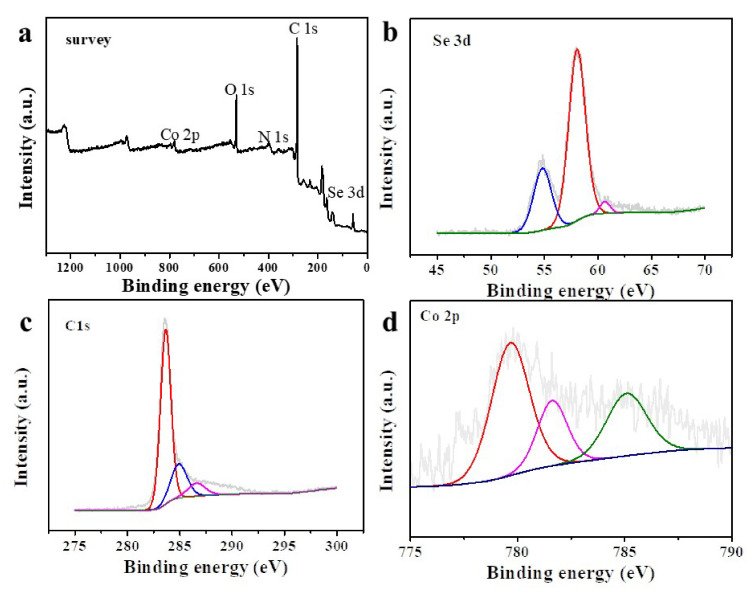
XPS full spectra of CoSe_2_-400 nanocomposite (**a**); high-resolution XPS spectrum of Se 3d (**b**), C 1s (**c**), and Co 2p (**d**).

**Figure 4 nanomaterials-12-00572-f004:**
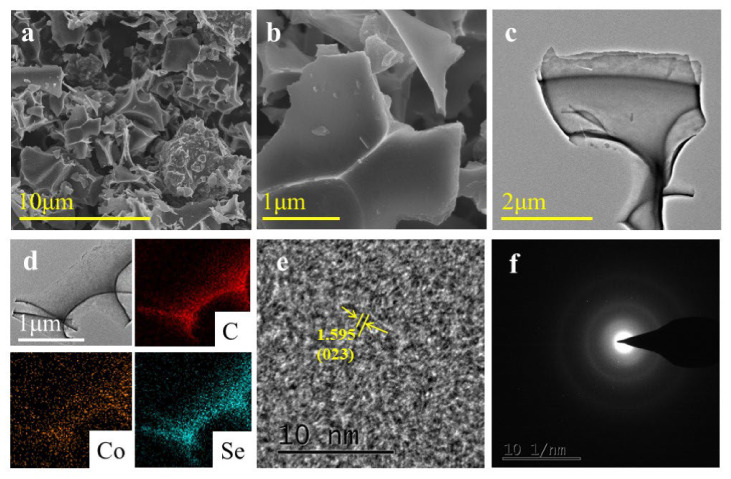
SEM images (**a**,**b**), TEM diagrams (**c**), EDX mapping images (**d**), HRTEM (**e**), and SAED images (**f**) of CoSe_2_−400, respectively.

**Figure 5 nanomaterials-12-00572-f005:**
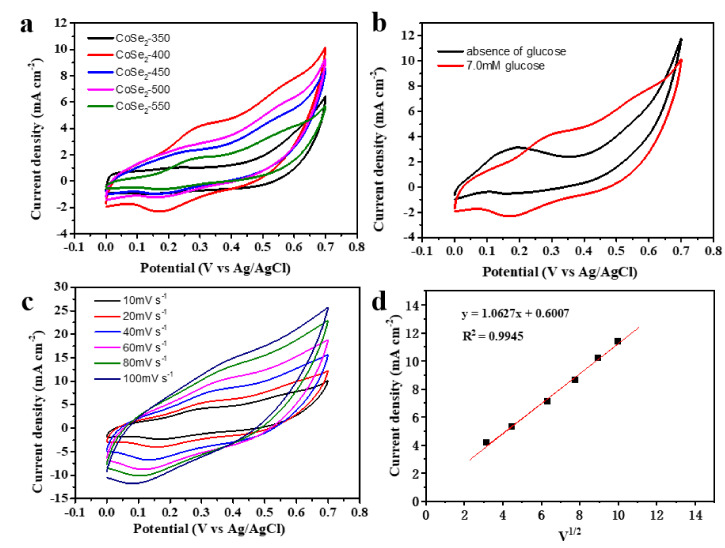
(**a**) CV curves of CoSe_2_-350, CoSe_2_-400, CoSe_2_-450, and CoSe_2_-500, CoSe_2_-550 composites. (**b**) CV curves of CoSe_2_-400 with or without glucose. (**c**) CV curves of CoSe_2_-400 at different scanning speeds. (**d**) Fitting curves of peak current density and square root of scanning rate corresponding to Figure 5c. The solution is aqueous 0.1 M KOH with 7.0 mM glucose.

**Figure 6 nanomaterials-12-00572-f006:**
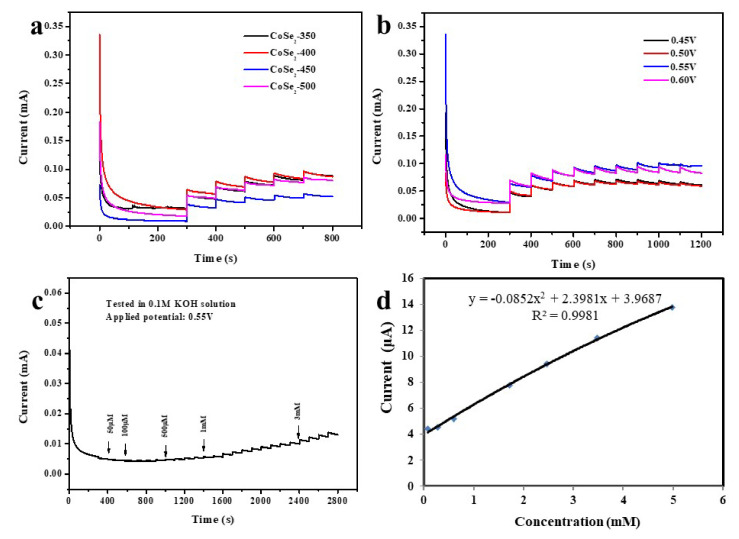
(**a**) I–t curves of CoSe_2_-350, CoSe_2_-400, CoSe_2_-450, and CoSe_2_-500 at 0.55 V applied voltage. (**b**) I–t curves at different applied voltages of CoSe_2_-400 nanocomposite; the solution was 0.1 M KOH electrolyte with 0.5 M glucose. (**c**) I–t curves of continuous glucose addition under 0.55 V applied voltage. (**d**) I–c linear fitting curve corresponding to continuous glucose addition.

**Figure 7 nanomaterials-12-00572-f007:**
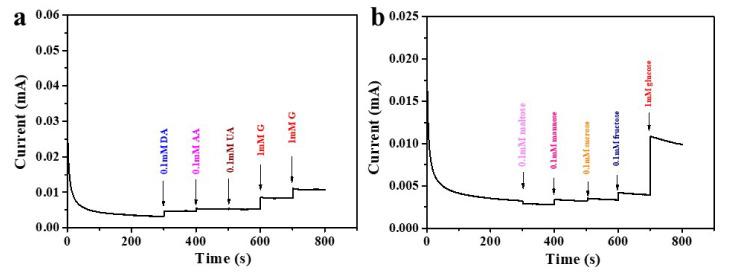
I–t curves of CoSe_2_-400 nanocomposite with the addition of 0.1 mM DA, AA, UA, and 1 mM glucose (**a**); 0.1 mM maltose, sucrose, mannose, fructose, and 1 mM glucose (**b**) in 0.1 M KOH electrolyte at 0.55 V applied voltage.

## Data Availability

Not applicable.

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
