# Peer review of "The Construct CoSe2 on Carbon Nanosheets as High Sensitivity Catalysts for Electro-Catalytic Oxidation of Glucose"

_nanomaterials, 2022, doi:10.3390/nano12030572_

Round 1
Reviewer 1 Report
Title: The Construct CoSe2 on Carbon Nanosheets by Low State Tem-2 plate Method for Electro-Catalytic Oxidation of Glucose
Recommendation: Major revision
- Authors must revise this current title. Its not effective and not suitable.
- Authors must add some more points about impact of the "glucose sensor via electrochemical method"
- Experimental section: The experiment with the synthesis procedure should be improved such as volume, concentration, time, temperature. The authors mentioned microliter without concentration. Its not suitable for the repeatability of the work. authors must check and revise.
- Elemental quantification of the materials is missing.
- Authors must improve Figure 1. (synthesis scheme)
- Scale bar is missing in Figure 4d.
- Authors should check all figures and caption in the manuscript.
- Error bar of the calibration plot is missing in Figure 5d, 6d.
- Authors must improve selectivity aspect. its not a good, current form of the experiment data.
- There are many grammatical and typographical errors. Please check the manuscript and refine it carefully.
Author Response
Dr. Editor and Reviewers,
We sincerely thank you very much for your help during the reviewing process on our manuscript titled “The Construct CoSe2 on Carbon Nanosheets by Low State Template Method for Electro-Catalytic Oxidation of Glucose”, Manuscript number nanomaterials-1563406. We also deeply appreciate the reviewers’ constructive suggestions, insightful criticisms, and valuable comments on our manuscript provided with a clear goal to make the manuscript better.
We carefully studied the reviewer’s comments and substantially revised our manuscript by taking into consideration of the reviewer’s recommendations. We believe that all issues raised by the reviewers have been adequately addressed and the manuscript has been greatly improved. Enclosed please find our replies to all comments. The following is a summary of our response to the reviewers’ comments and the corresponding changes.
Best regards,
Ying Chang, Ph.D
Reviewer #1:
Recommendation: Major revision
Comment 1. Authors must revise this current title. It’s not effective and not suitable.
Response: Thanks for the comment. We have revised the title in the manuscript.
The title “The Construct CoSe2 on Carbon Nanosheets by Low State Template Method for Electro-Catalytic Oxidation of Glucose” has been revised as “The Construct CoSe2 on Carbon Nanosheets as High Sensitivity Catalysts for Electro-Catalytic Oxidation of Glucose”.
Action: We have revised the title in the manuscript.
Comment 2. Authors must add some more points about impact of the "glucose sensor via electrochemical method".
Response: Thanks for the suggestion. We have now supplemented the impact of the "glucose sensor via electrochemical method" and cited related references in the manuscript.
Electrochemical glucose biosensors have drawn enormous attention with the advantages of simple operation, low detection line, and wide linear range. Therefore, glucose sensors exhibited broad application prospects in the fields of biological, clinical diagnosis, medicine, environmental protection, and military medicine.
Related references:
- W.-C. Lee, K.-B. Kim, N. G. Gurudatt, K. K. Hussain, C. S. Choi, D.-S. Park and Y.-B. Shim, Biosen. Bioelectron., 2019, 130, 48-54.
- C. Tang, N. Cheng, Z. Pu, W. Xing, X. Sun, Angew. Chem. Int. Ed., 2015, 54, 9351-9355.
Action: We have supplemented the impact of the "glucose sensor via electrochemical method" and cited related references in the manuscript.
Comment 3. Experimental section: The experiment with the synthesis procedure should be improved such as volume, concentration, time, temperature. The authors mentioned microliter without concentration. It’s not suitable for the repeatability of the work. authors must check and revise.
Response: Thanks for the comment. We have carefully supplemented the experiment details. The concentration of glucose (0.06 M), NaCl (1.2 M) have revised. And we have revised the experimental details as “Pipette 6 μL (0.02 M) slurry onto the electrode surface”.
Action: We have carefully supplemented the experiment details in the manuscript.
Comment 4. Elemental quantification of the materials is missing.
Response: Thanks for the comment. In fact, we have conducted the EDS characterization to clarify elemental quantification in Supplementary Figure 1. As shown in the table (inset), Co element accounts for 2.2%, and Se element accounts for 13.18% in our nanocomposite. And we have added the message in the text.
Action: We have showed the EDS characterization to clarify elemental quantification in Supplementary Figure 1.
Comment 5. Authors must improve Figure 1. (synthesis scheme)
Response: Thanks for the comment. We have updated Figure 1 carefully to look clearer and more scientific.
Action: We have updated Figure 1 in the manuscript.
Comment 6. Scale bar is missing in Figure 4d.
Response: Thanks for the comment. We have supplemented the scale bar in Figure 4d and updated Figure 4.
Action: We have supplemented the scale bar in Figure 4d and updated Figure 4.
Comment 7. Authors should check all figures and caption in the manuscript.
Response: Thanks for the comment. We have checked all figures and revised some captions in detail.
We have checked the figures over and over again to ensure their correctness, and updated Figures 1, 4, 5 to make the figures more correct and beautiful. The caption of Figure 1 has revised to “Synthesis diagram of CoSe2 nanocomposites”; and the caption of Figure 3 has revised to “XPS full spectra of CoSe2-400 nanocomposite (a), high-resolution XPS spectrum of Se 3d (b), C 1s (c), and Co 2p (d)”; “The solution is aqueous 0.1 M KOH with 7.0 mM glucose” was added in the caption of Figure 5; the Figure 6a, b captions have correct as “(a) I-t curves of different samples CoSe2-350, CoSe2-400, CoSe2-450, CoSe2-500 at 0.55 V applied voltage, (b) I-t curves at different applied voltages of CoSe2-400 nanocomposite, the solution was 0.1M KOH electrolyte with 0.5 M glucose”. The caption of Figure 1 has revised to “I-t curves of CoSe2-400 nanocomposite with the addition of measured by adding 0.1 mM DA, AA, UA, and 1 mM glucose (a); 0.1 mM maltose, sucrose, mannose, fructose, and 1 mM glucose (b) in 0.1 M KOH electrolyte at 0.55 V applied voltage.”
Comment 8. Error bar of the calibration plot is missing in Figure 5d, 6d.
Response: Thanks for the comment. The Error bar is a classic way, but we think the current data can explain the results in the Figure 5d, 6d of this MS. The similar expression is shown in the other references (Journal of Materials Chemistry A 2019, 6(29): 14367-14379. Biosensors & Bioelectronics 2019, 130: 48-54.).
Comment 9. Authors must improve selectivity aspect. It’s not a good, current form of the experiment data.
Response: Thanks for the comment. The samples have been synthesized and tested for many times, and the data results and discussion are scientific. Higher selectivity and device testing are being carried out in other articles. We believe that better results will be found in the new research system.
Comment 10. There are many grammatical and typographical errors. Please check the manuscript and refine it carefully.
Response: Thanks for the reviewer’s comment. We have revised the manuscript thoroughly and tried to avoid any grammar or syntax errors as well as colloquial terms to meet the high criteria of nanomaterials. We replenished information with highlighted yellow, deleted unused statements with red fonts as shown in copy manuscript.
Action:
Dr. Editor and Reviewers,
We sincerely thank you very much for your help during the reviewing process on our manuscript titled “The Construct CoSe2 on Carbon Nanosheets by Low State Template Method for Electro-Catalytic Oxidation of Glucose”, Manuscript number nanomaterials-1563406. We also deeply appreciate the reviewers’ constructive suggestions, insightful criticisms, and valuable comments on our manuscript provided with a clear goal to make the manuscript better.
We carefully studied the reviewer’s comments and substantially revised our manuscript by taking into consideration of the reviewer’s recommendations. We believe that all issues raised by the reviewers have been adequately addressed and the manuscript has been greatly improved. Enclosed please find our replies to all comments. The following is a summary of our response to the reviewers’ comments and the corresponding changes.
Best regards,
Ying Chang, Ph.D
Reviewer #1:
Recommendation: Major revision
Comment 1. Authors must revise this current title. It’s not effective and not suitable.
Response: Thanks for the comment. We have revised the title in the manuscript.
The title “The Construct CoSe2 on Carbon Nanosheets by Low State Template Method for Electro-Catalytic Oxidation of Glucose” has been revised as “The Construct CoSe2 on Carbon Nanosheets as High Sensitivity Catalysts for Electro-Catalytic Oxidation of Glucose”.
Action: We have revised the title in the manuscript.
Comment 2. Authors must add some more points about impact of the "glucose sensor via electrochemical method".
Response: Thanks for the suggestion. We have now supplemented the impact of the "glucose sensor via electrochemical method" and cited related references in the manuscript.
Electrochemical glucose biosensors have drawn enormous attention with the advantages of simple operation, low detection line, and wide linear range. Therefore, glucose sensors exhibited broad application prospects in the fields of biological, clinical diagnosis, medicine, environmental protection, and military medicine.
Related references:
- W.-C. Lee, K.-B. Kim, N. G. Gurudatt, K. K. Hussain, C. S. Choi, D.-S. Park and Y.-B. Shim, Biosen. Bioelectron., 2019, 130, 48-54.
- C. Tang, N. Cheng, Z. Pu, W. Xing, X. Sun, Angew. Chem. Int. Ed., 2015, 54, 9351-9355.
Action: We have supplemented the impact of the "glucose sensor via electrochemical method" and cited related references in the manuscript.
Comment 3. Experimental section: The experiment with the synthesis procedure should be improved such as volume, concentration, time, temperature. The authors mentioned microliter without concentration. It’s not suitable for the repeatability of the work. authors must check and revise.
Response: Thanks for the comment. We have carefully supplemented the experiment details. The concentration of glucose (0.06 M), NaCl (1.2 M) have revised. And we have revised the experimental details as “Pipette 6 μL (0.02 M) slurry onto the electrode surface”.
Action: We have carefully supplemented the experiment details in the manuscript.
Comment 4. Elemental quantification of the materials is missing.
Response: Thanks for the comment. In fact, we have conducted the EDS characterization to clarify elemental quantification in Supplementary Figure 1. As shown in the table (inset), Co element accounts for 2.2%, and Se element accounts for 13.18% in our nanocomposite. And we have added the message in the text.
Action: We have showed the EDS characterization to clarify elemental quantification in Supplementary Figure 1.
Comment 5. Authors must improve Figure 1. (synthesis scheme)
Response: Thanks for the comment. We have updated Figure 1 carefully to look clearer and more scientific.
Action: We have updated Figure 1 in the manuscript.
Comment 6. Scale bar is missing in Figure 4d.
Response: Thanks for the comment. We have supplemented the scale bar in Figure 4d and updated Figure 4.
Action: We have supplemented the scale bar in Figure 4d and updated Figure 4.
Comment 7. Authors should check all figures and caption in the manuscript.
Response: Thanks for the comment. We have checked all figures and revised some captions in detail.
We have checked the figures over and over again to ensure their correctness, and updated Figures 1, 4, 5 to make the figures more correct and beautiful. The caption of Figure 1 has revised to “Synthesis diagram of CoSe2 nanocomposites”; and the caption of Figure 3 has revised to “XPS full spectra of CoSe2-400 nanocomposite (a), high-resolution XPS spectrum of Se 3d (b), C 1s (c), and Co 2p (d)”; “The solution is aqueous 0.1 M KOH with 7.0 mM glucose” was added in the caption of Figure 5; the Figure 6a, b captions have correct as “(a) I-t curves of different samples CoSe2-350, CoSe2-400, CoSe2-450, CoSe2-500 at 0.55 V applied voltage, (b) I-t curves at different applied voltages of CoSe2-400 nanocomposite, the solution was 0.1M KOH electrolyte with 0.5 M glucose”. The caption of Figure 1 has revised to “I-t curves of CoSe2-400 nanocomposite with the addition of measured by adding 0.1 mM DA, AA, UA, and 1 mM glucose (a); 0.1 mM maltose, sucrose, mannose, fructose, and 1 mM glucose (b) in 0.1 M KOH electrolyte at 0.55 V applied voltage.”
Comment 8. Error bar of the calibration plot is missing in Figure 5d, 6d.
Response: Thanks for the comment. The Error bar is a classic way, but we think the current data can explain the results in the Figure 5d, 6d of this MS. The similar expression is shown in the other references (Journal of Materials Chemistry A 2019, 6(29): 14367-14379. Biosensors & Bioelectronics 2019, 130: 48-54.).
Comment 9. Authors must improve selectivity aspect. It’s not a good, current form of the experiment data.
Response: Thanks for the comment. The samples have been synthesized and tested for many times, and the data results and discussion are scientific. Higher selectivity and device testing are being carried out in other articles. We believe that better results will be found in the new research system.
Comment 10. There are many grammatical and typographical errors. Please check the manuscript and refine it carefully.
Response: Thanks for the reviewer’s comment. We have revised the manuscript thoroughly and tried to avoid any grammar or syntax errors as well as colloquial terms to meet the high criteria of nanomaterials. We replenished information with highlighted yellow, deleted unused statements with red fonts as shown in copy manuscript.
Action: We have thoroughly revised the manuscript to avoid any grammar or syntax errors as well as colloquial terms. We replenished information with highlighted yellow, deleted unused statements with red fonts as shown in copy manuscript.
We have thoroughly revised the manuscript to avoid any grammar or syntax errors as well as colloquial terms. We replenished information with highlighted yellow, deleted unused statements with red fonts as shown in copy manuscript.

Reviewer 2 Report
Dear Editor,
The authors described here the development of a glucose sensor base on COSe2 for the electro-catalytic oxidation of glucose. The authors show the nanomaterial characterization by different techniques, presenting satisfactory results. The authors apply the material to fabricate a glucose sensor obtaining sufficient results. The experiments seem to be well performed; the manuscript is professionally written and shows interesting results for the applications of the synthesized nanomaterial. However, in the section corresponding to glucose sensor characterization, there are points that need to be addressed before considering the acceptance of the manuscript for publication at Nanomaterials.
- The term biosensor is used for the analytical device that uses biological recognition elements to detect and quantify a chemical, in the case of the present manuscript there is not a biological recognition element, the reviewer suggests changing the term biosensor to sensor.
- At page 2, line 50, could the authors include references to support the information.
- At Figures 5a, 5c and 5d, could the authors clarify if they performed the experiment only in KOH, or if the media already includes glucose. According to their text, the media is only KOH, if that is the case, could the authors explain why in figure 5b the shape of the figure in black is completely different from the CV’s shape of figures 5 a, c, and d? If the experiment is in presence of glucose, include the used concentration clarify in the text and figure caption.
- In figure 5d, is not clear the information that the authors would like to show, could be possible that the authors show only the linear regression and coefficient regression?
- Is figure 6a correct? If the authors already chose the electrode CoSe2 at 400 for having the best performance at voltammetry, why do the authors show i-t experiments with all the sensors? In this figure, 6a is difficult to observe which sensor has better sensitivity. Additionally, the authors mention that they performed additions on glucose 0.5 M in KOH, but the authors did not provide information about the volume of the cell or the final glucose concentration for each addition. Please clarify.
- In figure 6b, is difficult to observe the authors claim, the background current seems to be displaced according to the applied potential, and apparently, the response could be similar, probably the authors cloud analyses better if they plot current vs concentration and compare sensitivities for each potential applied.
- In the text corresponding to figure 6 d, the authors claim to have a linear fitting up to 5mM, however in figure 6d, the authors show a polynomial fitting, could the authors explain?
The reviewer recommended minor revision before considering the acceptance of the manuscript for publishing at Nanomaterials.
Sincerely,
The reviewer.
Author Response
Dr. Editor and Reviewers,
We sincerely thank you very much for your help during the reviewing process on our manuscript titled “The Construct CoSe2 on Carbon Nanosheets by Low State Template Method for Electro-Catalytic Oxidation of Glucose”, Manuscript number nanomaterials-1563406. We also deeply appreciate the reviewers’ constructive suggestions, insightful criticisms, and valuable comments on our manuscript provided with a clear goal to make the manuscript better.
We carefully studied the reviewer’s comments and substantially revised our manuscript by taking into consideration of the reviewer’s recommendations. We believe that all issues raised by the reviewers have been adequately addressed and the manuscript has been greatly improved. Enclosed please find our replies to all comments. The following is a summary of our response to the reviewers’ comments and the corresponding changes.
Best regards,
Ying Chang, Ph.D
Reviewer #2:
The authors described here the development of a glucose sensor base on COSe2 for the electro-catalytic oxidation of glucose. The authors show the nanomaterial characterization by different techniques, presenting satisfactory results. The authors apply the material to fabricate a glucose sensor obtaining sufficient results. The experiments seem to be well performed; the manuscript is professionally written and shows interesting results for the applications of the synthesized nanomaterial. However, in the section corresponding to glucose sensor characterization, there are points that need to be addressed before considering the acceptance of the manuscript for publication at Nanomaterials.
Comment 1. The term biosensor is used for the analytical device that uses biological recognition elements to detect and quantify a chemical, in the case of the present manuscript there is not a biological recognition element, the reviewer suggests changing the term biosensor to sensor.
Response: Thanks for the suggestion. We agree with the reviewer’s suggestion, and have replaced all the “biosensor” into “sensor” in the manuscript.
Action: We have replaced all the “biosensor” into “sensor” in the manuscript.
Comment 2. At page 2, line 50, could the authors include references to support the information.
Response: Thanks for the suggestion. Actually, we did omit the corresponding references. Related references have supplemented in the text and the order of references has also been reordered.
- Z. Wang, J. Li, X. Tian, X. Wang, Y. Yu, K.A. Owusu, L. He, L. Mai, ACS Appl. Mater. Interfaces, 2016, 8, 19386-19392.
- A.T. Swesi, J. Masud, M. Nath, Energy Environ. Sci., 2016, 9, 1771-1782.
- C. Tang, N. Cheng, Z. Pu, W. Xing, X. Sun, Angew. Chem. Int. Ed., 2015, 54, 9351-9355.
Action: We have supplemented the references in the manuscript and the order of references has been reordered.
Comment 3. At Figures 5a, 5c and 5d, could the authors clarify if they performed the experiment only in KOH, or if the media already includes glucose. According to their text, the media is only KOH, if that is the case, could the authors explain why in figure 5b the shape of the figure in black is completely different from the CV’s shape of figures 5 a, c, and d? If the experiment is in presence of glucose, include the used concentration clarify in the text and figure caption.
Response: Thanks for the comment. We have checked the figures in detail.
In figure 5b, the black curve was conducted in the absence of glucose, and other curves in figure 5 a, b, c, d were measured in the solution with 7.0 mM glucose. We showed the condition in the figures but maybe not clearly. We have supplemented the experiment details in the text and figure caption. “Set the scanning potential range of 0~0.7 V (vs Ag/AgCl) in 0.1 M KOH solution” have corrected as “Set the scanning potential range of 0~0.7 V (vs Ag/AgCl) in 0.1 M KOH solution with 7.0 mM glucose” in the text; and in the figure caption, we added the experiment details “The solution is aqueous 0.1 M KOH with 7.0 mM glucose”.
Action: We have checked and supplemented the experiment details in the text and figure caption.
Comment 4. In figure 5d, is not clear the information that the authors would like to show, could be possible that the authors show only the linear regression and coefficient regression?
Response: Thanks for the suggestion. We have updated Figure 5d to make the message clearer. We have deleted the unclear inset and supplemented the linear regression and coefficient regression. As shown in Figure 6d, the linear regression was: y = 1.0627x + 0.6007, and the coefficient regression was 1.0627.
Action: We have updated Figure 5d and supplemented the linear regression and coefficient regression.
Comment 5. Is figure 6a correct? If the authors already chose the electrode CoSe2 at 400 for having the best performance at voltammetry, why do the authors show i-t experiments with all the sensors? In this figure, 6a is difficult to observe which sensor has better sensitivity. Additionally, the authors mention that they performed additions on glucose 0.5 M in KOH, but the authors did not provide information about the volume of the cell or the final glucose concentration for each addition. Please clarify.
Response: Thanks for the comment.
For this work, our original intention was to compare CoSe2 nanomaterials at different synthetic methods and select the high-performance one. From the CV curves, the CoSe2-400 did show better response than others, but to confirm the performance of a material, it should be characterized from multiple performance tests. Figure 6a illustrated that CoSe2-400 composite also exhibited better performance under the I-t test, which confirmed the excellent characteristics of this material. So we conducted this experiment.
As you mentioned, in Figure 6a, it is not clear to confirm which sensor completely possesses better sensitivity. But our experimental data is repeated multiple times, ensuring data authenticity, CoSe2-400 materials did show better performance.
In I-t tests, we added 0.5 M glucose into 0.1 M KOH electrolyte for each time. In Figure 6a, we only did the test with the addition of glucose for 5 times, so the final glucose concentration is 0.25 M, the final glucose concentration can be calculated by the specific experimental steps. Besides, we supplemented the experimental operation in the Figure 6 caption to express clearer results.
Action: We have clarified the question above and supplemented the experimental operation in the Figure 6 caption to express clearer results.
Comment 6. In figure 6b, is difficult to observe the authors claim, the background current seems to be displaced according to the applied potential, and apparently, the response could be similar, probably the authors cloud analyses better if they plot current vs concentration and compare sensitivities for each potential applied.
Response: Thanks for the comment. To obtain optimal amperometric response to glucose, the effect of different applied potentials on the response current of the CoSe2-400 was also investigated. The figure 6b shows the amperometric responses of the electrode to successive injections of glucose at various applied potentials. The oxidation current of glucose at the CoSe2-400 increased with the increase of the potential (0.45 V to-0.60 V). An applied potential of 0.55 V led to a remarkable enhancement in the current response upon each addition of glucose which is show in a larger image (Figure S2). By comparison, the optimal potential (0.50 V) was chosen as the working potential.
Figure S2. I-t curves at different applied voltages of CoSe2-400 nanocomposite, the current at 200s is as the backgroud.
Action: We have carefully revised the manuscript in the manuscript.
Comment 7. In the text corresponding to figure 6 d, the authors claim to have a linear fitting up to 5mM, however in figure 6d, the authors show a polynomial fitting, could the authors explain?
Response: Thanks for the comment. We indeed made a fault in the language expression with “linear fitting curve”, we have revised the expression as “fitting curve”, but the accuracy of data is guaranteed.
Action: We have checked the data carefully and corrected the expression as accurately as possible.

Round 2
Reviewer 1 Report
Revised form of the manuscript has acceptable for publication.
Author Response
Thanks for the reviewer’s comment. We have revised the manuscript thoroughly and tried to avoid any grammar or syntax errors as well as colloquial terms to meet the high criteria of nanomaterials.
